# Extending the Anion Channelrhodopsin-Based Toolbox for Plant Optogenetics

**DOI:** 10.3390/membranes11040287

**Published:** 2021-04-14

**Authors:** Yang Zhou, Meiqi Ding, Xiaodong Duan, Kai R. Konrad, Georg Nagel, Shiqiang Gao

**Affiliations:** 1Institute of Physiology, Department of Neurophysiology, Biocenter, University of Wuerzburg, 97070 Wuerzburg, Germany; yang.zhou@stud-mail.uni-wuerzburg.de (Y.Z.); duanxd@sustech.edu.cn (X.D.); georg.nagel@botanik.uni-wuerzburg.de (G.N.); 2Institute for Molecular Plant Physiology and Biophysics, Julius-von-Sachs-Institute, Biocenter, University of Wuerzburg, 97082 Wuerzburg, Germany; meiqi.ding@stud-mail.uni-wuerzburg.de (M.D.); kai.korad@botanik.uni-wuerzburg.de (K.R.K.); 3Department of Biology, College of Science, Southern University of Science and Technology (SUSTech), Shenzhen 518055, China

**Keywords:** optogenetics, rhodopsin, light-sensitive anion channel, surface potential recording, pollen tube

## Abstract

Optogenetics was developed in the field of neuroscience and is most commonly using light-sensitive rhodopsins to control the neural activities. Lately, we have expanded this technique into plant science by co-expression of a chloroplast-targeted β-carotene dioxygenase and an improved anion channelrhodopsin *Gt*ACR1 from the green alga *Guillardia theta*. The growth of *Nicotiana tabacum* pollen tube can then be manipulated by localized green light illumination. To extend the application of analogous optogenetic tools in the pollen tube system, we engineered another two ACRs, *Gt*ACR2, and ZipACR, which have different action spectra, light sensitivity and kinetic features, and characterized them in *Xenopus laevis* oocytes, *Nicotiana benthamiana* leaves and *N. tabacum* pollen tubes. We found that the similar molecular engineering method used to improve *Gt*ACR1 also enhanced *Gt*ACR2 and ZipACR performance in *Xenopus laevis* oocytes. The ZipACR1 performed in *N. benthamiana* mesophyll cells and *N. tabacum* pollen tubes with faster kinetics and reduced light sensitivity, allowing for optogenetic control of anion fluxes with better temporal resolution. The reduced light sensitivity would potentially facilitate future application in plants, grown under low ambient white light, combined with an optogenetic manipulation triggered by stronger green light.

## 1. Introduction

Optogenetics employs rhodopsins or other photoreceptors to manipulate defined events in biological systems with high spatial and temporal resolution [1,2]. The most commonly used optogenetic tools are the microbial rhodopsins with light-gated ion channel or pump functions [3]. The discovery of Channelrhodopsin-2 (ChR2), a light-sensitive cation channel from the unicellular green alga *Chlamydomonas reinhardtii*, paved the way for optogenetic applications [4]. The light-sensitive cation channel ChR2 can be used to depolarize neurons and trigger action potential spiking [5,6,7,8,9]. The light-sensitive anion channelrhodopsin *Gt*ACR1 from the green algae *Guillardia theta* was discovered later [10] and enabled electric silencing in certain neurons via hyperpolarizing the plasma membrane [11,12,13,14]. With the unique advantages of fine spatial and temporal resolution, microbial opsin-based optogenetic tools advanced neuroscience research greatly [1,15].

Optogenetics can be potentially applied in plant physiology to study plant ion and electrical signaling. In recent years, several strategies were put into effort to develop optogenetic techniques in higher plants. For example, the red- and far-red-light regulated PhyB-PIF6 system [16], the green-light sensitive CarH-VP16 system [17] and a “PULSE” tool [18] were employed to control gene expression in higher plants. A synthetic blue-light gated K^+^ channel “BLINK1” was recently used to control the activity of stomata and plant growth [19].

However, in contrast to the widely used microbial rhodopsin-based optogenetic tools employed in animal cell systems, the widespread application in plants has so far failed to emerge. A recent attempt was using ChR2-XXL, harboring a point mutation in ChR2 [20], to depolarize the membrane potential in *Arabidopsis thaliana* [21]. In this study, the leaf discs or epidermal strips expressing ChR2-XXL had to be incubated in bath solutions containing all-trans-retinal, the essential co-factor and chromophore for microbial opsins [22], to reconstitute a functional ChR2-XXL. This method is, however, less applicable for investigations in entire living plants. To overcome this limitation and improve the application of microbial rhodopsins in plants, we recently introduced a chloroplast-targeted β-carotene dioxygenase to produce all-trans-retinal from the abundant β-carotene pool in chloroplasts and plastids. By co-expression of the dioxygenase with an improved anion channelrhodopsin ACR1 2.0, we have successfully manipulated the membrane potential of plant cells through anion redistribution and we were able to guide *N. tabacum* pollen tube growth with green light [23].

The synthesis of the chromophore retinal in the whole plant, which is not otherwise available, represents a breakthrough for using rhodopsins in plants [24]. As demonstrated in our recent case study, it enabled the use of ACR1 2.0 as a versatile optogenetic tool [23] which likely initiated the application of a variety of other microbial rhodopsins in plant optogenetic research approaches. With the aim to expand the application of ACRs in plant systems, we engineered and expressed other ACRs such as *Gt*ACR2 (ACR2 for short) from *Guillardia theta* [10] and ZipACR from *Proteomonas sulcata* [25]. The ACR2 has a blue-shifted action spectrum while ZipACR has faster kinetics and a reduced light sensitivity [10,25]. In this study, we confirmed that both ACR2 and ZipACR could be applied to plant cells by a similar N-terminal modification strategy used for ACR1 [23]. Both ACRs could be functionally expressed in the plasma membrane and depolarize the membrane potential while ZipACR1 was proven suitable for faster anion transport kinetics. The reduced light sensitivity of ZipACR could provide for advanced easy to apply optogenetic approaches because it might allow for plant growth under low white light regime while additional stronger green light application enables optogenetic manipulation.

## 2. Materials and Methods

### 2.1. Plasmid Construction

Anion channelrhodopsins *Gt*ACR1, *Gt*ACR2, and ZipACR were synthesized by Invitrogen GeneArt Gene Synthesis according to the published sequence (accession numbers KP171708, KP171709 and KX879679, respectively). The genes expressed in oocytes are cloned into the pGEM-HE vector. The synthetic signal peptide LucyRho (LR for short) [26] was fused to the N-terminal of ACRs. The 5′-plasma membrane trafficking signal (T) and a 3’- ER export signal (E) derived from Kir2.1 [27] were fused to the C-terminal of the ACRs to improve the expression and plasma membrane targeting. eYFP is inserted between T and E to provide for a semi-quantitative measure for opsin expression level as the expression marker.

For transient expression in *N. benthamiana* leaves, a binary vector pCAMBIA-3300 harboring a UBQ10 promoter is used. The 2.0 versions of ACRs are fused to the C-terminal of retinal production gene “Ret” with a P2A self-cleaving linker derived from porcine teschovirus-1 [28]. For the expression in *N. tabacum* pollen tubes, the whole insert cassette was transferred from the pCAMBIA-3300 vector to a pSAT-based vector with a UBQ10 promoter.

The constructed plasmids were transformed into *Escherichia coli* for amplification. The plasmid extraction is performed using the QIAprep or Spin MiniPrep 250 Kit (QIAGEN) and PureLinkTM HiPure Plasmid Filter Midiprep Kit (Invitrogen by Thermo Fisher Scientific, Roskilde, Denmark) following the manufacturer’s instructions. All constructs were verified by commercially available DNA sequencing (Eurofins Genomics Germany GmbH).

### 2.2. Two-Electrode Voltage-Clamp (TEVC) in Xenopus laevis Oocytes

Complementary RNA (cRNA) from ACR1, ACR1 2.0, ACR2, ACR2 2.0, ZipACR, and ZipACR 2.0 were synthesized by in vitro transcription using the AmpliCap-MaxT7 High Yield Message Maker Kit (Epicentre Biotechnologies, Madison, WI, USA), solubilized in nuclease-free water and stored at −20 °C.

Total of 20 ng cRNA of these ACRs were injected into *Xenopus laevis* oocytes, and the injected oocytes were incubated in ND96 solution (96 mM NaCl, 2 mM KCl, 1 mM CaCl_2_, 1 mM MgCl_2_, 10 mM HEPES, pH 7.4) at 16 °C for 2–3 days. The photocurrents were recorded with a two-electrode voltage-clamp amplifier (TURBO TEC-03X, NPI electronic GmbH, Tamm, Germany) at room temperature (~25 °C) in standard Ringer’s solution (110 mM NaCl, 5 mM KCl, 2 mM CaCl_2_, 1 mM MgCl_2_, 10 mM HEPES, pH 7.4). Electrode capillaries (Φ = 1.5 99 mm, Wall thickness 0.178 mm, Hilgenberg, Malsfeld, Germany) with tip resistance of 0.4 –1 MΩ were filled with 3 M KCl for the impalement of oocytes. A USB-6221 DAQ interface (National Instruments, Austin, TX, USA) and WinWCP V5.3.4 software (University of Strathclyde, Glasgow, UK) were used for data acquisition.

The 532 nm and 473 nm laser (Changchun New Industries Optoelectronics Tech, Changchun, China) were used for illumination with light intensities of ~100 µW/mm^2^. The light intensities were measured with a Plus 2 power & energy meter (Laserpoint, Milan, Italy).

### 2.3. Transient Expression in N. benthamiana Leaves by Agro-Infiltration

The plasmids for *N. benthamiana* leaf expression were transferred into the *Agrobacterium tumefaciens* strain GV3101 by an electroporation protocol [29]. Positive colonies were selected on lysogeny broth (LB)-agar plates with 100 μg/mL kanamycin, 25 μg/mL gentamycin, and 10 μg/mL rifampicin at 28 °C and confirmed by PCR.

Following the protocol of Li et al. [30], the confirmed positive *A. tumefaciens* was cultured overnight in LB medium (including 150 μM acetosyringone) at 28 °C. The cultured *A. tumefaciens* was centrifuged for 10 min at 4500 rpm. The sediment was washed thrice with an infiltration buffer (10 mM MgCl_2_, 10 mM pH 5.6 MES-K, 150 μM acetosyringone) and then was adjusted to OD_600_ = ~0.3–0.4 with the infiltration buffer. The resuspended bacteria were then infiltrated into the leaves of 4–6-week-old *N. benthamiana* plants through the abaxial epidermis via a 1 mL syringe. 3 days post infiltration, protein expression was verified by the eYFP fluorescence under a stereo-microscope.

### 2.4. Surface Potential Recordings with N. benthamiana Leaves

The intact *N. benthamiana* plants expressing respective ACRs were prepared according to the reported method [31]. 24 h before the measurement, the plants were watered to keep the soil moisture. The plant leaves were mounted stable and a drop of 10 mM KCl solution containing 0.5% (wt/vol) agar was placed on their surface to establish electric conductivity. The diameter of the round-shaped drop was kept between 2–6 mm. Ag/AgCl wires were connected to the microelectrode amplifier (Axon geneclamp 500) and were immersed into the KCl/agar drop without touching the leaf surface. The reference electrode was put into a 10 µL pipette tip filled with KCl/agar solution and inserted in the soil of the pot. Light application was performed with a 532 nm and a 473 nm laser (Changchun New Industries Optoelectronics Tech, Changchun, China) at ~180 μW/mm^2^ on the recording spot of the leaf. Data were digitalized using a NA USB-6221 interface (National Instruments) and recorded with the WinWCP V5.3.4 software (University of Strathclyde, Glassglow, UK).

### 2.5. Transient Expression in N. tabacum Pollen Tubes

The transient expression of pSAT vector-based plasmids into *N. tabacum* pollen tubes was performed by particle bombardment of pollen grains according to the previously reported method [32]. For each construct, 2.5 mg tungsten particles (Bio-Rad, 1.1 µm diameter) were coated with 10 µg plasmid DNA and were shot into hydrated *N. tabacum* pollen grains which were spread on the surface of 1% agar plates containing germination medium as previously described [32]. The pollen germination medium, with 420 mOsm kg-1 osmolalities, consisted of 1 mM MES, 1.6 mM H_3_BO_3_ and 0.4 mM CaCl_2_. The osmolality was adjusted using D (+)–sucrose and measured by Vapor Pressure Osmometer 5520. The pH of the medium was adjusted to 5.8 with Tris.

After bombardment, pollen grains were suspended with germination medium and mixed with 40 °C warm germination medium containing low melt agarose (Carl Roth, final agarose concentration 1%). Finally, the mixture was transferred to recording chambers with a glass coverslip bottom pre-coated with 0.01% poly-Lysine (Sigma-Aldrich, Søborg, Denmark) and incubated at 25 °C (in the dark) for 5 h. The pollen tubes expressing respective proteins were selected by eYFP fluorescence.

### 2.6. Light-Induced Voltage and Current Recording in Pollen Tubes

Voltage-clamp and current-clamp experiments with tobacco pollen tubes were performed as previously described [33]. Double-barreled microelectrodes filled with 300 mM KCl were connected to the voltage-clamp amplifier (TEC-05X; NPI Electronic, Tamm, Germany) by Ag/AgCl wires. The amplifier was equipped with head stages of ≥1013 Ω input impedance. Reference electrodes filled with 300 mM KCl were plugged with 2% agar containing 300 mM KCl. Double-barreled electrodes were impaled about 100 µm behind the pollen tube tip through a piezo-driven micro-manipulator (Sensapex). WinWCP V5.3.4 software (University of Strathclyde, Glassglow, UK) was used to apply the voltage-clamp or current-clamp protocols. Photocurrents were recorded at different holding potentials ranging from −160 mV to +40 mV (∆20 mV increments) with the 532 nm and 473 nm laser (Changchun New Industries Optoelectronics Tech, Changchun, China) at 3.5 mW/mm^2^ and 4 mW/mm^2^ light intensities, respectively.

### 2.7. Confocal Images Processing

To screen for expression and image subcellular localization of ACRs in different cells, images were taken with a confocal laser scanning microscope (Leica SP5, Leica Microsystems CMS, Mannheim, Germany). eYFP fluorescence emission was captured between 520 and 580 nm after excitation at 496 nm. YFP fluorescence was monitored in *N. benthamiana* leaves and *N. tabacum* pollen tubes with a dipping 25× HCX IRAPO 925/0.95 and 40× water immersion HC PL FLUOTAR 910/0.3 objective. For the oocyte observation, a dipping 25× HCX IRAPO 925/0.95 was used. ImageJ software was used for image processing.

### 2.8. Data Analysis

The fluorescence images were processed with ImageJ software. The line chart and histogram were drawn by Graghpad 8.0.2 or Origin 2018 software. The kinetics of ACRs in oocytes was analyzed by Clampfit 10.7. Significance between two groups was determined by Student’s T-Test. For more than two groups, the significance was analyzed by One way ANOVA with IBM SPSS Statistics 26. For the labelling in the figures: *, *p* ≤ 0.05; **, *p* ≤ 0.01; ***, *p* ≤ 0.001.

## 3. Results

### 3.1. Characterization of Engineered ACRs in Xenopus laevis Oocytes

After functional expression of ACR1 in plant cells by simultaneous all-trans-retinal synthesis [23], we aim to test additional anion channelrhodopsins (Figure 1a) to expand the channelrhodopsin-based plant optogenetic toolkit with different characteristics for specific plant applications. To improve the expression of ACR2 and ZipACR in the plasma membrane, the optimized construct ACR2 2.0 and ZipACR 2.0 versions were built by fusing the LR signal peptide to the N-terminal of ACR2 and ZipACR in addition to the plasma membrane trafficking signal (T) and ER export signal peptides (E) in the C-terminus (see the schemes in Figure 1b). In comparison to ACR2 and ZipACR, both engineered ACR2 2.0 and ZipACR 2.0 exhibited higher expression level in the plasma membrane of oocytes and resulted in improved photocurrents (Figure 1c,d), which is similar to ACR1 2.0 (Figure 1b–d). Consistent with a previous report [25], ZipACR 2.0 showed the fastest off kinetics among the three ACRs tested with a τ_off_ = 2.3 ± 0.5 ms (Figure 1f). Compared to ACR1 2.0, the ACR2 2.0 and ZipACR 2.0 exhibited a lower light sensitivity with the effective light power density for 50% photocurrent (EPD_50_) of 320 µW/mm^2^ and 830 µW/mm^2^, respectively.

### 3.2. Functional Expression of Different ACRs in N. benthamiana Leaves

The optimized ACR2 2.0 and ZipACR 2.0 anion channels were expressed together with the retinal producing enzyme MbDio “Ret” in a tandem construct with a P2A self-cleavage linker in between (see the schemes in Figure 2a). The fusion constructs will be co-translated, but as two separate proteins: Ret and ACR-YFP. Ret-ACR2 2.0 and Ret-ZipACR 2.0 were transiently expressed together with Ret-ACR1 2.0 as a positive control in *N. benthamiana* leaves after agro-infiltration for a functional comparison of the ACRs. Three days post infiltration, the expression of eYFP was confirmed by fluorescence imaging. The Ret-ACR1 2.0 showed strong expression and triggered robust electric responses (Figure 2b,c), similar to our previous report [23]. The Ret-ACR2 2.0 showed a strong expression judging from the YFP fluorescence but the blurred diffuse YFP signal suggested a high proportion of the proteins to localize within the cell or in endomembranes (Figure 2b). Thus a relatively weak electric response was detected in surface potential recordings after blue light (473 nm, close to the action spectrum maximum of ACR2 [10]) illuminations (Figure 2c), compatible with the action spectrum of ACR2 [10]. The fluorescence upon Ret-ZipACR 2.0 expression showed clear plasma membrane-targeted localization, in line with a stronger electric response in surface potential recordings after green light (532 nm) illumination (Figure 2b,c), which is close to the peak of the action spectrum of ZipACR [25]. The WT plants and Ret-eYFP-expressing control plants showed no response to the same blue or green light stimulation (Figure 2c,d). Taken these results together, Ret-ZipACR 2.0 expressed in *N. benthamiana* leaves displayed a well-defined light-triggered electric response and could be employed as a versatile optogenetic tool in plant studies when fast on-off kinetics is required (Figure 2e). On the contrary, Ret-ACR2 2.0 is less well suited as a useful optogenetic tool in plants due to its undefined localization and little impact on the membrane potential as indicated by the weak surface potential response (Figure 2e).

In comparison to Ret-ACR1 2.0, the light sensitivity of Ret-ZipACR 2.0 is significantly lower (Figure 2f,g). 18 µW/mm^2^ green light is not able to trigger obvious membrane potentials changes with Ret-ZipACR 2.0, when intracellular electrodes are employed (Figure 2f). Nonetheless, 50 µW/mm^2^ green light can trigger recognizable membrane potential changes (Figure 2f) with Ret-ZipACR 2.0. This suggests that transgenic Ret-ZipACR 2.0 plants would most likely grow under low ambient white light without substantial optogenetic stimulation. For example, the white light intensity in greenhouses is often below or around 50 µW/mm^2^ and the green light fraction of ambient white light should not activate ZipACR 2.0, however, optogenetic manipulation could then be triggered by additional green light of higher intensity.

### 3.3. Functional Expression of ACR2 and ZipACR in N. tabacum Pollen Tubes

The ACR2 2.0 and ZipACR 2.0 were further tested in *N. tabacum* pollen tubes upon transient expression after particle bombardment of pollen grains. Functional characterization was performed by means of fluorescence imaging and electrophysiological recordings, in 5 h grown pollen tubes. In contrast to the clear plasma membrane YFP signal in Ret-ACR1 2.0 expressing pollen tubes, Ret-ACR2 2.0 and Ret-ZipACR 2.0 displayed a less specific plasma membrane signal, but additionally a diffuse intracellular YFP signal (Figure 3a). In line with their plasma membrane localization, in both Ret-ACR2 2.0 and Ret-ZipACR 2.0 expressing pollen tubes, blue light or green light application resulted in significant photocurrents that are clearly separated from the endogenous currents before and after light stimulation (Figure 3b,c). The photocurrents in Ret-ZipACR 2.0 expressing pollen tubes were approximately twice as high as that of Ret-ACR2 2.0 pollen tubes (Figure 3b,c), most likely due to a better plasma membrane targeting.

To better elucidate the performance, applicability, and differences between ACRs in plant cells, we further utilized the voltage-clamp measurements in pollen tubes. In line with the oocyte data (Figure 1f), the closing kinetics of Ret-ZipACR 2.0 was faster than that of Ret-ACR1 2.0 (Figure 3d), but the values obtained in pollen tubes were quite different from those obtained from *Xenopus* oocytes. While in oocytes the Ret-ZipACR1 2.0 off-kinetics was more than 200 times faster than those for Ret-ACR1 2.0 (Figure 1f) it was only two times faster in tobacco pollen tubes (Figure 3d). It is quite unlikely that the off-kinetics of Ret-ACR1 2.0 and Ret-ZipACR 2.0 was influenced by the activity of endogenous channels, because amplitude of the native currents and their deactivation kinetics are negligibly small in relation to the light-induced photocurrents (Figure 3b). The faster deactivation kinetics of Ret-ZipACR 2.0 compared to Ret-ACR1 2.0 can be visualized particularly well in a direct comparison of the voltage changes induced in pollen tubes. Upon optogenetic stimulation with repetitive 50 ms lasting 3.5 mW/mm^2^ green light (532 nm) pulses of 5 Hz, Ret-ZipACR 2.0 triggered fully reversible voltage changes (Figure 4). The Ret-ACR1 2.0 induced voltage changes, by contrast, did not recover to the pre-stimulus level when light-triggering was applied with the same frequency (Figure 4).

## 4. Discussion

A breakthrough in the use of opsins as optogenetic tools in plants was the recently established method for the production of retinal in plants [23]. It facilitated the functional expression of microbial opsins in plants, and in a case study, demonstrated ACR1 2.0 driven anion efflux to control the growth direction of pollen tubes [23]. In the current study, we have explored the possibility of using additional light-gated anion channels for plant research, namely ACR2 and ZipACR. We characterized ACR2 and ZipACR and compared them to the well-characterized ACR1 in *Xenopus* oocytes first. Subsequently, we extended the functional comparison of the light-gated ACRs to *N. benthamiana* mesophyll cells and *N. tabacum* pollen tubes. ACR2 was reported to have similar light-induced photocurrent amplitude with ACR1 but faster kinetics and a blue-shifted action spectrum [10]. ZipACR was described to generate large current amplitudes and a much faster photocycle than the previously reported ACRs [25]. In cultured mouse hippocampal neurons, ZipACR could inhibit neuronal spiking precisely at up to 50 Hz frequency [25].

We applied in this study a similar molecular engineering strategy as recently described [23], to improve or enable the application of opsins in plants in the first place. We added the synthetic signal peptide LR in the N-terminal, which improved the expression, consequently increasing the photocurrents of ACR2 and ZipACR in *Xenopus* oocytes dramatically (Figure 1). However, when expressing Ret-ACR2 2.0 in the two plant cell systems, good expression, but weak plasma membrane localization was associated with comparably smaller photocurrent and light induced surface potential changes in *N. tabacum* pollen tubes and *N. benthamiana* mesophyll cells, respectively (Figure 2 and Figure 3). In contrast to Ret-ACR2 2.0, Ret-ZipACR 2.0 was characterized by far better plasma membrane localization and displayed a much stronger light-triggered electric response (Figure 2 and Figure 3). We were able to confirm the known faster on-off kinetics of ZipACR in contrast to ACR1 in *Xenopus* oocytes as well as in tobacco mesophyll cells and pollen tubes (Figure 2 and Figure 3). However, we could demonstrate that the photocurrent deactivation kinetics of ZipACR was only twice as fast as the one of ACR1 in plant cells (Figure 3). The faster gating of ZipACR might represent an advantage in some plant applications and will facilitate tighter temporal optogenetic manipulation. Another property that could be advantageous for some applications in plants is the lower light sensitivity of Ret-ZipACR 2.0 with respect to Ret-ACR1 (Figure 2). This could probably allow transgenic plants, carrying Ret-ZipACR 2.0, to grow under low ambient white light, e.g., in the greenhouse, where the optogenetic manipulation can then be triggered by addition of stronger green light.

## 5. Conclusions

In summary, we found the same molecular engineering strategies used to express ACR1 successfully in plants to apply to other anion channelrhodopsin. In this study, we identified ZipACR as a suitable and possible advantageous ACR to perform plant optogenetic manipulation with faster kinetics. The weaker light-sensitivity of ZipACR in comparison to ACR1 might allow unaffected growth under low light conditions. Based on the results of this study one important point needs to be pointed out. It is crucial to confirm rhodopsin function in plant expression systems like *N. benthamiana* mesophyll cells and *N. tabacum* pollen tubes before the time-consuming generation of transgene plants. We show for example that in *Xenopus* oocytes ACR2 2.0 performed well, but not in plant cells. ACR2 2.0 was not correctly plasma membrane targeted and subsequently did not generate robust electric responses like ACR1 2.0 or ZipACR 2.0. The increasing availability of a multitude of opsins from all kingdoms will in the future enable to dissect plant signaling pathways in a novel and non-invasive manner.

## Figures and Tables

**Figure 1 membranes-11-00287-f001:**
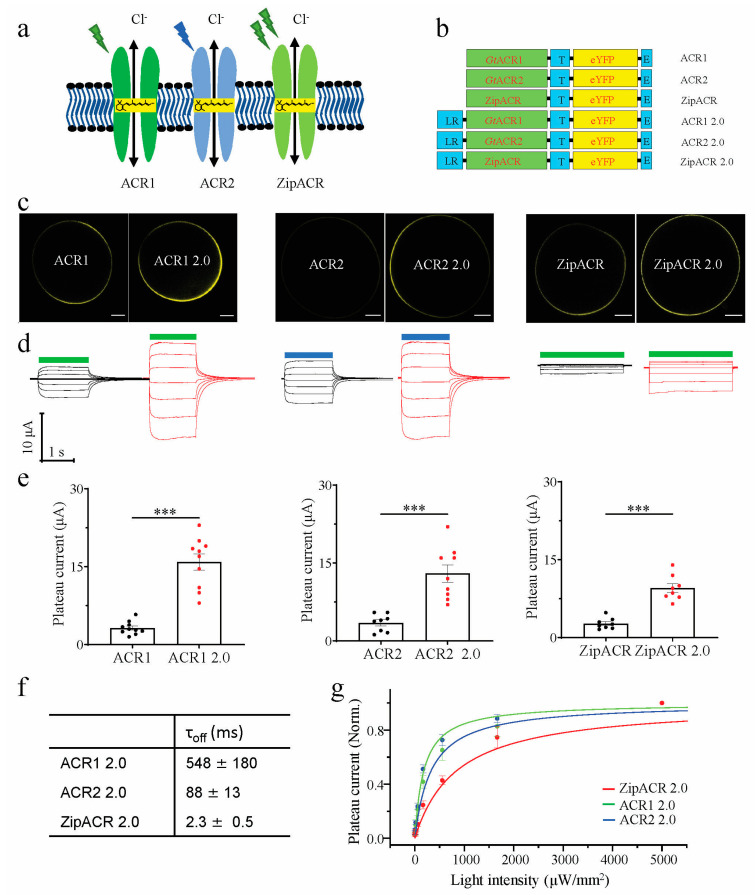
Characterization of engineered ACRs in *Xenopus laevis* oocytes. (**a**) Schemes of the anion channelrhodopsins conducting chloride ions; the chromophore all-trans-retinal is indicated in the yellow frame. ACR1 is green light-activated, ACR2 is blue light-activated, and ZipACR needs green light of higher qunatity to be activated. (**b**) Scheme of the ACR1, ACR1 2.0, ACR2, ACR2 2.0, ZipACR, and ZipACR 2.0 DNA constructs, LR: the LucyRho membrane targeting signal, T: plasma membrane trafficking signal, E: ER export signal peptides. (**c**) Confocal fluorescence images of oocytes expressing YFP-fusions of ACR1, ACR1 2.0, ACR2, ACR2 2.0, ZipACR, and ZipACR 2.0 (n = 3), scale bar = 200 μm. (**d**) Representative photocurrent traces of ACR1, ACR1 2.0, ACR2, ACR2 2.0, ZipACR, and ZipACR 2.0 measured by two-electrode voltage-clamp (TEVC) in oocytes. Holding potentials during voltage-clamp pulse protocols were from −90 mV to 30 mV (from bottom to top) in 20 mV increments, green bar represents green light (532 nm, 100 µW/mm^2^) illumination and blue bar represent blue light (473 nm, 100 µW/mm^2^) illumination. (**e**) Comparison of the light-induced photocurrents of ACR1(n = 10), ACR1 2.0 (n = 10), ACR2 (n = 8), ACR2 2.0 (n = 8), ZipACR (n = 8), and ZipACR 2.0 (n = 8) in oocytes, the statistical significance was analyzed by Student’s T-Test, ACR1 versus ACR1 2.0, *p* = 2.4 × 10^−6^, ACR2 versus ACR2 2.0, *p* = 1.3 × 10^−4^, ZipACR versus ZipACR 2.0, *p* = 4.0 × 10^−6^, Error bars = s.e.m. (**f**) Channel closing kinetics (τ_off_) was determined by a single exponential fit with Clampfit 10.7, for ACR1 2.0, τ_off_ = 548 ± 180 ms, for ACR2 2.0, τ_off_ = 88 ± 13 ms, for ZipACR 2.0, τ_off_ = 2.3 ± 0.5 ms. Data were obtained from measurements at −70 mV and shown as mean ± s.e.m, n = 3. (**g**) Comparison of the light sensitivity of ACR1 2.0 (n = 6), ACR2 2.0 (n = 5) and ZipACR 2.0 (n = 5). Error bars = s.e.m. For ACR1 2.0, EPD_50_ = ~180 μW/mm^2^, for ACR2 2.0, EPD_50_ = ~320 μW/mm^2^, and for ZipACR 2.0, EPD_50_ = ~830 μW/mm^2^. For each cell, the data points at different light intensities were divided by the photocurrent recorded under the highest light intensity (5000 μW/mm^2^) for normalization. ***, *p* ≤ 0.001.

**Figure 2 membranes-11-00287-f002:**
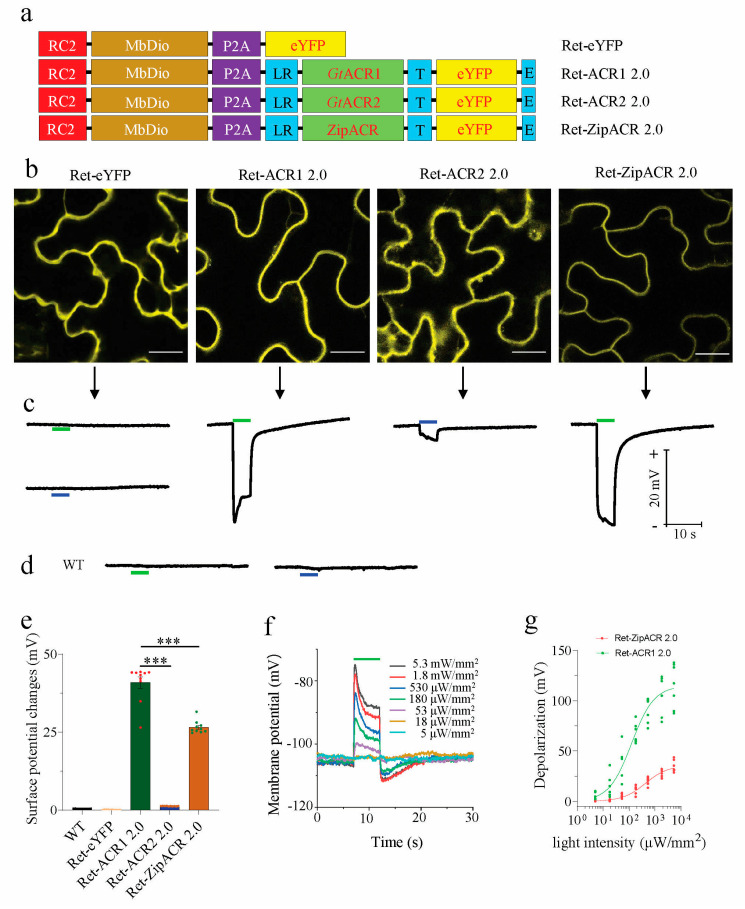
Functional expression of ACR2 and ZipACR in *N. benthamiana* leaves. (**a**) Schemes of the Ret-eYFP, Ret-ACR1 2.0, Ret-ACR2 2.0 and Ret-ZipACR 2.0 DNA constructs. (**b**) Representative confocal images of *N benthamiana* epidermal cells expressing Ret-eYFP (n =15), Ret-ACR1 2.0 (n = 15), Ret-ACR2 2.0 (n = 11) and Ret-ZipACR 2.0 (n = 3), the images were taken 3 days post infiltration (dpi), scale bar = 200 μm. (**c**) Surface potential recordings on *N. benthamiana* leaves expressing Ret-eYFP, Ret-ACR1 2.0, Ret-ACR2 2.0 and Ret-ZipACR 2.0 upon 532 nm green light (green bar) or 473 nm blue light (blue bar) with 180 μW/mm^2^ for 5 s, n = 10 for each construct, and the recording was performed 3 dpi. (**d**) The surface potential recordings on the leaves of wild-type *N. benthamiana* (WT) with 180 μW/mm^2^ 532 nm green light (green bar) and 473 nm blue light (blue bar) for 5 s, n = 10. (**e**) The comparison of light-induced surface potential changes on *N. benthamiana* leaves expressing Ret-eYFP, Ret-ACR1 2.0, Ret-ACR2 2.0, and Ret-ZipACR 2.0 illuminated with the same wavelength as in (**d**), WT and Ret-eYFP-expressing plants were exposed to green light (532 nm). The statistical significance is performed by One-way ANOVA following the Turkey test. Ret-ACR 1 2.0 vs. Ret-ACR 2 2.0, *p* = 6.8 × 10^−13^, Ret-ACR1 2.0 vs. Ret-ZipACR 2.0, *p* = 7.2 × 10^−13^. (**f**) A representative membrane potential recording in mesophyll cells of *N. benthamiana* leaves expressing Ret-ZipACR 2.0 under 5 s green light (532 nm, green bar) of different intensities. (**g**) The comparison of the light sensitivity of Ret-ACR1 2.0 and Ret-ZipACR 2.0, for Ret-ACR1, EPD_50_ = ~125 μW/mm^2^, and for Ret-ZipACR 2.0, EPD_50_ = ~420 μW/mm^2^). ***, *p* ≤ 0.001.

**Figure 3 membranes-11-00287-f003:**
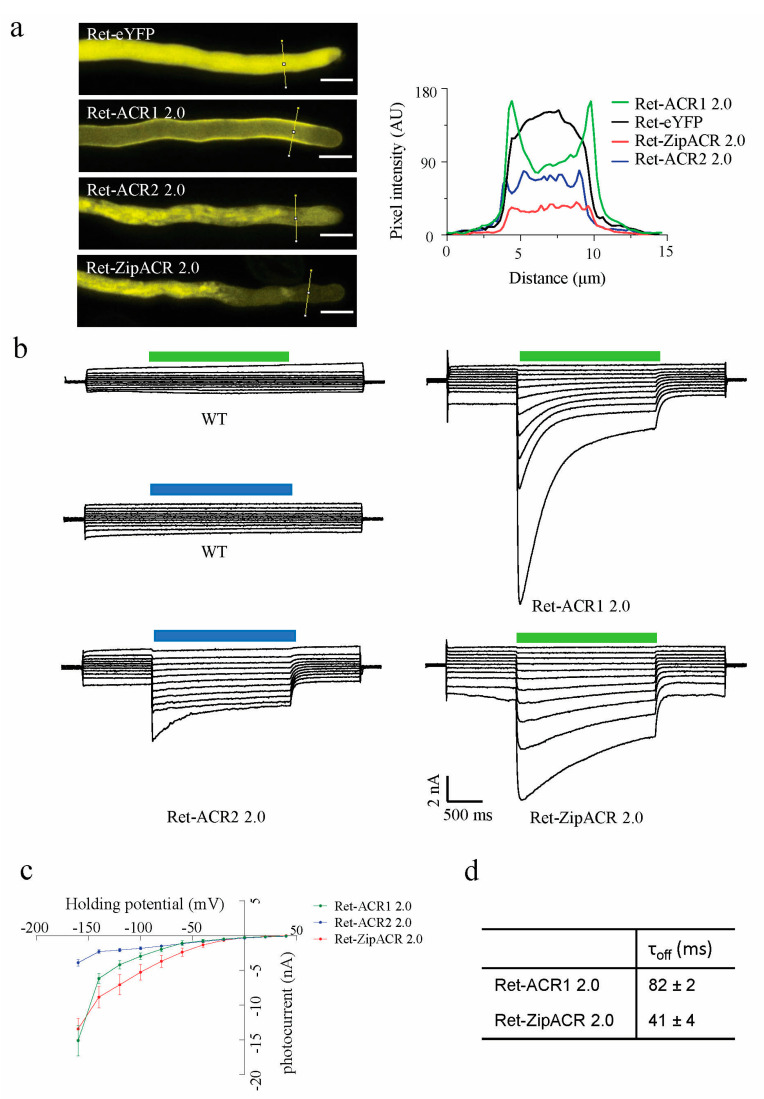
Functional expression of ACR2 and ZipACR in *N. tabacum* pollen tubes. (**a**) Representative confocal images of pollen tubes transiently expressing Ret-eYFP (n = 6), Ret-ACR1 2.0 (n = 6), Ret-ACR2 2.0 (n = 6) and Ret-ZipACR 2.0 (n = 6), the images were taken 5 h after transformation, scale bar = 10 μm. For the analysis on the pixel intensity of the images on the right side, the line segments were drawn across the pollen tube images (left) and analyzed by Plot Profile (ImageJ). The x-axis represents the distance along the lines and the y-axis is the pixel intensity. (**b**) Representative photocurrent traces of wild- type (WT) tobacco pollen tubes and pollen tubes transiently expressing Ret-eYFP, Ret-ACR1 2.0, Ret-ACR2 2.0 and Ret-ZipACR 2.0 (all under UBQ10 promoter) in voltage-clamp recordings. Green bar indicates 2 s of 3.5 mW/mm^2^ green light (532 nm), blue bar indicates 2 s of 4 mW/mm^2^ blue light (473 nm) pulse. (**c**), Current-voltage relation of photocurrents in pollen tubes transiently expressing Ret-ACR1 2.0, Ret-ACR2 2.0 and Ret-ZipACR 2.0. Ret-ACR1 2.0 and Ret-ZipACR 2.0 were illuminated with 3.5 mW/mm^2^ 532 nm light, and Ret-ACR2 2.0 is illuminated with 4 mW/mm^2^ 473 nm light. Error bar = s.e.m, n = 5 cells. (**d**) For Ret-ACR1 2.0, τ_off_ = 82.3 ± 2.21 ms. For ZipACR 2.0, τ_off_ = 40.7 ± 3.8 ms. Data were obtained from measurements at −80 mV and shown as mean ± s.e.m, n = 5.

**Figure 4 membranes-11-00287-f004:**
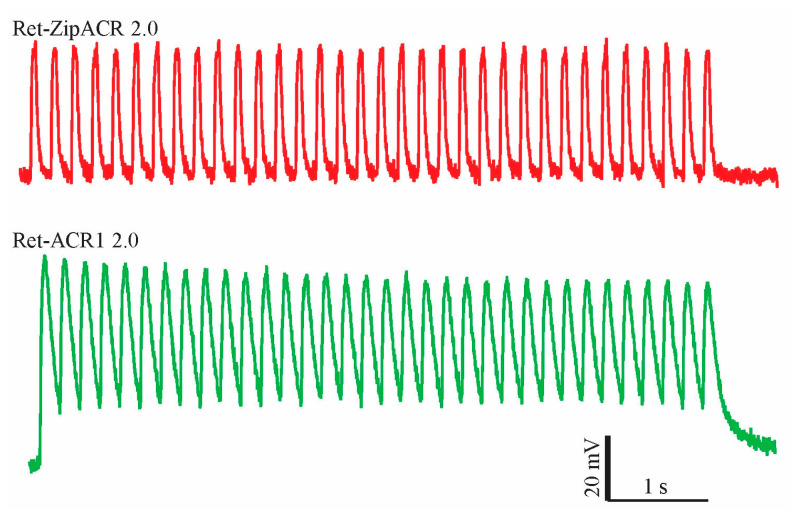
Light-induced depolarizations in tobacco pollen tubes transiently expressing Ret-ZipACR 2.0 (red line) or Ret-ACR1 2.0 (green line). The pollen tubes were illuminated by 50 ms light flashes of 3.5 mW/mm^2^ green light (532 nm) at 5 Hz, the downward deflections of the voltage traces correspond to the darkness-regime (150 ms).

## Data Availability

All relevant source data are provided as additional files. All DNA and plasmids are available from the corresponding authors upon reasonable requests.

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
