# Peer review of "Extending the Anion Channelrhodopsin-Based Toolbox for Plant Optogenetics"

_membranes, 2021, doi:10.3390/membranes11040287_

Round 1

Reviewer 1 Report

The manuscript by Zhou et al. entitled “Extending the anion channelrhodopsin-based toolbox for plant optogenetics” reports photoregulation of the membrane potential of Xenopus laevis oocytes, Nicotiana benthamiana leaf epidermal and mesophyll cells and N. tabacum pollen tubes upon expression of ACR2 from Guillardia theta and ZipACR from Proteomonas sulcata, two cryptophyte light-gated anion channels. This study is an extension of the earlier published work (Ref. 23) in which GtACR1, another G. theta ACR, was used for similar purposes. Cryptophyte ACRs have already been shown to control neuronal firing with light in many laboratories, but their application to plant systems has only begun. Therefore, this manuscript provides a valuable contribution to this new area of research. I find no major problems with it and recommend it for publication after a minor revision addressing the issues outlined below.

Minor issues:

Line 22: Change “timporal” to “temporal”.

Lines 48-49: “…in contrast to the widely used microbial rhodopsin-based optogenetic tools used in animal cell systems…”

Replace one of the “used” with a different word.

Line 65: Change “sulcate” to “sulcata” and “spectra” to “spectrum”.

Line 186: “…ZipACR 2.0 exhibited a lower light sensitivity…”

From Fig. 1C it looks like ZipACR and GtACR2 were expressed at lower levels than GtACR1. Was it taken into account when plotting the light intensity dependence? If not, it has to be explained. Also, please explain how plateau photocurrent was normalized in Fig. 1g.

Lines 209-210: “The Ret-ACR2 2.0 showed strong expression but diffuse YFP signal in the periphery of the cells…”

Please explain how strong expression was determined, if not by YFP fluorescence. If you mean that YFP fluorescence was observed not only in the plasma membrane, but also in the cytoplasm, please write so.

Line 328: “generation of transgene plants generation”

Please delete the second “generation”.

Line 330: Please replace “nice” with “robust”.

Author Response

Reviewer 1:

The manuscript by Zhou et al. entitled “Extending the anion channelrhodopsin-based toolbox for plant optogenetics” reports photoregulation of the membrane potential of Xenopus laevis oocytes, Nicotiana benthamiana leaf epidermal and mesophyll cells and N. tabacum pollen tubes upon expression of ACR2 from Guillardia theta and ZipACR from Proteomonas sulcata, two cryptophyte light-gated anion channels. This study is an extension of the earlier published work (Ref. 23) in which GtACR1, another G. theta ACR, was used for similar purposes. Cryptophyte ACRs have already been shown to control neuronal firing with light in many laboratories, but their application to plant systems has only begun. Therefore, this manuscript provides a valuable contribution to this new area of research. I find no major problems with it and recommend it for publication after a minor revision addressing the issues outlined below.

Minor issues:

Line 22: Change “timporal” to “temporal”.

Lines 48-49: “…in contrast to the widely used microbial rhodopsin-based optogenetic tools used in animal cell systems…”

Replace one of the “used” with a different word.

Line 65: Change “sulcate” to “sulcata” and “spectra” to “spectrum”.

Answer: We thank the reviewer for the careful proofreading, and we have changed the above 3 points as suggested.

Line 186: “…ZipACR 2.0 exhibited a lower light sensitivity…”

From Fig. 1C it looks like ZipACR and GtACR2 were expressed at lower levels than GtACR1. Was it taken into account when plotting the light intensity dependence? If not, it has to be explained. Also, please explain how plateau photocurrent was normalized in Fig. 1g.

Answer: The expression level will influence the absolute response strength. It is true that ZipACR and GtACR2 were expressed at lower levels than GtACR1 in Xenopus oocytes. However, the light sensitivity as a parameter to describe the photon capture ability of photoreceptors is not influenced by the expression level. We have added the explanation of normalization in the legend of Fig. 1g.

Lines 209-210: “The Ret-ACR2 2.0 showed strong expression but diffuse YFP signal in the periphery of the cells…”

Please explain how strong expression was determined, if not by YFP fluorescence. If you mean that YFP fluorescence was observed not only in the plasma membrane, but also in the cytoplasm, please write so.

Answer: Yes, the expression was determined by the YFP fluorescence. We have now modified the sentence to make it clear. “The Ret-ACR1 2.0 showed strong expression and triggered robust electric responses (fig. 2b, c), similar to our previous report [23]. The Ret-ACR2 2.0 showed a strong expression judging from the YFP fluorescence but the blurred diffuse YFP signal suggested a high proportion of the proteins to localize within the cell or in endomembranes”

Line 328: “generation of transgene plants generation”

Please delete the second “generation”.

Line 330: Please replace “nice” with “robust”.

Answer: We thank the reviewer for the careful proofreading, and we have changed the above 2 points as suggested.

Reviewer 2 Report

Nice paper that extends on recent work by the authors. Some English can be improved/typographical errors e.g. "temporal" ln32; sentence beginning in ln71 not clear; constructed plasmids "were" ln87; xxx ln 130; 2.6 "Light ln148. 

Minor comments:

Fig. 1a is not clear to me - I suggest having a cartoon for each ACR1 (green light); ACR2 (blue light); ZipACR (higher green light needed) - this will emphasise the difference in their properties and why they are being tested. ACR2 offers BL control whereas ZipACR offers a way to tune GL sensitivity. 

ln82 - expression levels here are assessed by YFP fluorescence. Please state to clarify. 

Section 3.2 More description on the fusion will help i.e. it's a co-translational fusion but it will be cleaved in vivo into separate Ret and ACR proteins. 

Also in this section, the ACR1 data is not referred to. 

A major conclusions of the paper it that ZipACR holds promise as a tool in plants under ambient white light conditions. I invite the authors to test this with the tobacco system. Leaves can be exposed to moderate white light (which should not stimulate ZipACR) and then exposed to supplemental GL to induce activity. This will help support one of the main conclusions of the paper.

Figure 3a - it's hard to see plasma membrane localisation - can the authors quantify the fluorescence through a trans section - this will define the localisation more

The photo cycle kinetics of ZipACR (and ACR1) differ when examined in pollen. The authors discuss this but what is not clear is whether the advantage of ZipACR still holds (lower sensitivity). Can this be tested to again further support the utility of this anion channel tool in different plant cell types?

Author Response

Reviewer 2:

Nice paper that extends on recent work by the authors. Some English can be improved/typographical errors e.g. "temporal" ln32; sentence beginning in ln71 not clear; constructed plasmids "were" ln87; xxx ln 130; 2.6 "Light ln148. 

Answer: We thank the reviewer for the careful proofreading, and we have changed the above points as suggested.

Minor comments:

Fig. 1a is not clear to me - I suggest having a cartoon for each ACR1 (green light); ACR2 (blue light); ZipACR (higher green light needed) - this will emphasise the difference in their properties and why they are being tested. ACR2 offers BL control whereas ZipACR offers a way to tune GL sensitivity. 

Answer: This is a very good idea to improve the visibility. We have modified the Fig. 1a as suggested.

ln82 - expression levels here are assessed by YFP fluorescence. Please state to clarify. 

Answer: Clarified as suggested.

Section 3.2 More description on the fusion will help i.e. it's a co-translational fusion but it will be cleaved in vivo into separate Ret and ACR proteins. 

Answer: We have added more description as suggested.

Also in this section, the ACR1 data is not referred to. 

Answer: We have added more description about ACR1 as suggested.

A major conclusions of the paper it that ZipACR holds promise as a tool in plants under ambient white light conditions. I invite the authors to test this with the tobacco system. Leaves can be exposed to moderate white light (which should not stimulate ZipACR) and then exposed to supplemental GL to induce activity. This will help support one of the main conclusions of the paper.

Answer: We agree with the reviewer that a final test with transgene tobacco or other plant will be very meaningful. However, in this study, we focus on the transient systems to confirm previous strategy and screen new candidates for future plant transgene. We are preparing the transgene plants and will definitely test this in the future. But the transgene work will require another half years’ time and we would like to keep it as a follow up of the current study.

Figure 3a - it's hard to see plasma membrane localisation - can the authors quantify the fluorescence through a trans section - this will define the localisation more

Answer: This is a very good idea to improve the visibility. We have added a trans section pixel analysis as suggested. (See the revised Figure 3a)

The photo cycle kinetics of ZipACR (and ACR1) differ when examined in pollen. The authors discuss this but what is not clear is whether the advantage of ZipACR still holds (lower sensitivity). Can this be tested to again further support the utility of this anion channel tool in different plant cell types?

Answer: We agree with the reviewer that the light sensitivity of ZipACR might change in pollen. However, the light sensitivity of ACRs in pollen tubes seem not important for the plant growth from our previous study. We have expressed Ret-ACR1 under the pollen-specific LeLAT promoter and  plant growth is uneffected, they flower and produce seeds normally although the ACR1 is highly sensitive to light. The probable reason is that tobacco can bloom at night and finish pollination before being influenced by strong sunlight. The tobacco flower also tend to close at day time to avoid strong white light. For other plants, even the ZipACR is very sensitive to light in the pollens and the flower is open in day time, the flower can also be easily covered and protected from light because pollen germination normally do not need light.
